

# Effects of physical activity on anxiety levels in college students: mediating role of emotion regulation

Xinxin Sheng[1], Xili Wen[2], Jiangshan Liu[1], Xiuxiu Zhou[3] and Kai Li[2]

[1] Institutes of Physical Education, Changzhou University, Changzhou, Jiangsu, China
[2] School of Physical Education, Shanghai University of Sport, Shanghai, China
[3] People's Liberation Army Unit 63613, Gansu, Lanzhou, China

## ABSTRACT

**Objectives.** To explore the effects of physical activity on anxiety levels in college students, as well as to examine the mediating role of emotion regulation.

**Methods.** A convenience sample of 1,721 college students from Shanghai, Jiangsu, Shandong, Guangxi, and Hunan was used to conduct an evaluation and a survey through the Physical Activity Rating Scale (PARS-3), the Anxiety Self-Rating Scale (SAS), and the Emotion Regulation Scale (ERQ).

**Results.** College students' anxiety level, cognitive reappraisal, and expression inhibition scores were $(44.72 \pm 10.37)$, $(30.16 \pm 6.51)$, and $(16.96 \pm 4.99)$, respectively. There were significant grade and physical activity level differences in anxiety levels and cognitive reappraisal, and significant gender and physical activity level differences in expression inhibition among college students. Process model 4 mediated effect regression analysis showed that physical activity had a significant positive effect on cognitive reappraisal ($R^2 = 0.14$, $\beta = 0.04$, $P < 0.001$), and physical activity did not have a significant expression inhibition effect ($R^2 = 0.17$, $\beta = 0.01$, $P = 0.27$). Physical activity ($\beta = -0.03$, $P = 0.012$), cognitive reappraisal ($\beta = -0.59$, $P < 0.001$), and expression inhibition had a significant effect on ($\beta = 0.57$, $P < 0.001$) anxiety levels ($R^2 = 0.37$). In the model effect relationship, the direct and indirect effects of physical activity on anxiety levels were $-0.028$ and $-0.019$, respectively.

**Conclusion.** Physical activity has a significant negative effect on college students' anxiety levels. Cognitive reappraisal is a mediating variable for the effect of physical activity on anxiety levels. The higher the level of physical activity and the higher the intensity of the activity, the lower the level of anxiety.

Corresponding author
Kai Li, kai.li@sus.edu.cn

## INTRODUCTION

Anxiety is an emotional state that includes inner unease, unpleasantness, worry, and a sense of fear about anticipated events. It is often accompanied by neurotic behaviors, such as pacing back and forth, physical discomfort, and rumination (*Chand & Marwaha, 2024*). The psychological problems of young college students in China are serious, and are mainly reflected in anxiety, obsession, depression, interpersonal sensitivity, hostility, paranoia,

and insomnia, among others (*Chen, Zhang & Yu, 2022*; *Wang & Xu, 2021*). In addition, anxiety is one of the most common psychological problems among college students, as it is prone to contribute to excessive psychological pressure and distress, and falls into a state of "mental sub-health". Research shows that the detection rate of anxiety, one of mental health problems, has increased from 0.66% to 82.5% over the past 10 years. The anxiety level of college students has been increasing significantly (*Chen, Zhang & Yu, 2022*). According to the results of a study on the mental health of college students, 42% of college students experienced varying degrees of anxiety in the face of the spread of the epidemic (*Han, 2020*). About 20.6% of Chinese college students have anxiety symptoms (*Zhan et al., 2021*). During the epidemic, 18.2% of college students suffered from anxiety symptoms (*Han et al., 2021*). College students are in an important stage of transition from adolescence to adulthood and are in a critical period of cognitive, emotional, social, and psychological development. Therefore, thinking about their future career and life is the main cause of anxiety among college students. With the complication of social factors, the challenges and pressures faced by current college students are increasing day by day. College students are gradually becoming a group at higher risk of feeling anxious, which means if they are not regulated by appropriate methods, anxiety will affect their normal life and study, and even seriously affect the development of their personality. Therefore, exploring effective ways to ease anxiety is an urgent research topic for us at present.

Numerous studies have shown that physical activity plays an important role in mental health (*Jiang, Zhang & Mao, 2018*). Mental health problems may develop due to lack of physical activity (*Tajik et al., 2017*). Anxiety is one of the main problems affecting mental health, and physical activity can distract attention from anxiety, can relieve anxiety to a certain extent, and have an ameliorative and therapeutic effect on anxiety (*Anderson & Brice, 2011*; *Ruby et al., 2011*). Physical activities has a significant intervention effect on anxiety (*Wipfli, Rethorst & Landers, 2008*). From a physiological mechanism perspective, the beneficial effects of regular exercise are typically attributed to the production of endorphins, which help regulate mood and enhance the function of the prefrontal cortex (*Daniela et al., 2022*). In addition, recent research has found that emotional regulation is an important factor in mental health (*Cloitre et al., 2019*), such as reducing the anger caused by arguments, sadness caused by the death of families, stress caused by exams and other negative emotions. The role of emotion regulation is particularly important. Does emotion regulation play an important role in the relationship between physical activity and anxiety? Against such background, this study aims to explore how physical activity affects anxiety levels and the potential influencing mechanisms. It also supplements the theoretical basis for enhancing individual mental health, aiming to provide effective physical exercise methods and applicable emotion regulation strategies to reduce the level and duration of anxiety among college students. It offers insights for developing corresponding practical intervention goals, thereby better equipping students to cope with life's challenges and adversities, and to prevent and reduce the occurrence of negative events.

## Relationship between physical activity and anxiety

Physical activity is an activity with a certain intensity, frequency, and duration for the purpose of improving physical health, with physical activity as content and means (*Li, 2017*). The *Guidelines on Physical Activity and Sedentary Behavior (2020)* issued by WHO clearly suggests the recommendation to enhance physical activity (*Bull et al., 2020*). Empirical studies have also demonstrated that anxiety levels can be effectively lowered through regular and moderate physical activity (*Carek, Laibstain & Carek, 2011*; *Dong et al., 2022*; *Li et al., 2023*). Facilitating physical activity is a health promotion approach to preventing or treating anxiety disorders (*Kandola & Stubbs, 2020*). Additionally, studies have found that physical activity with different levels and frequency have different effects on anxiety. High-intensity exercise relieves anxiety more than low-intensity exercise does, and those who exercise more frequently having fewer anxiety symptoms compared to those who exercise less frequently (*Ji et al., 2022*). Therefore, the existing studies suggest that physical activity negatively predicts anxiety levels.

## Mediating role of emotion regulation

Emotion regulation is the process by which individuals regulate their emotions based on the environment they are in, and the way people change their emotional responses (*Niu et al., 2023*). It consists of extrinsic and intrinsic processes that are responsible for detecting, assessing, and changing emotional responses (*Thompson, 1994*). Emotion regulation is a multidimensional structure that includes different strategies of emotion regulation, such as reappraisal, expressive inhibition, distraction, and avoidance (*Cisler & Olatunji, 2012*). Emotion regulation is related to general cognitive abilities and involves brain regions in the prefrontal cortex associated with executive functions. It downregulates the responses of emotion-related areas, including the amygdala (*Mohammed, Kosonogov & Lyusin, 2022*; *Steward et al., 2021*). Inadequate emotion regulation has been proved to be an important cause of anxiety and depression (*Wirtz et al., 2014*). College students who regularly do physical activity show stronger emotion regulation skills (*Xue & Zhao, 2018*), and emotion regulation is an important antecedent variable that college students commit to physical activity, and it has a significant effect. Therefore, improving the emotion regulation ability of college students can effectively improve the physical activity engagement status (*Zhu, Lu & Dong, 2016*) Emotion regulation plays a mediating role in the relationship between physical activity and mental health (*Jiang, Zhang & Mao, 2018*). Moreover, physical activity helps to cultivate positive emotions and deepen the effect of emotion regulation while relieving college students' anxiety (*Xu & Du, 2021*). Psychological problems can also be alleviated indirectly by influencing the use of emotion regulation strategies (*Xu, Zhang & Yang, 2022*).

## Objectives and hypotheses of the study

To conclude, although there have been studies that have elaborated the relationship between physical activity and anxiety, physical activity and emotion regulation, and emotion regulation and anxiety from their respective viewpoints at present, very few of them have explored the relationship between physical activity, anxiety, and emotion regulation. There

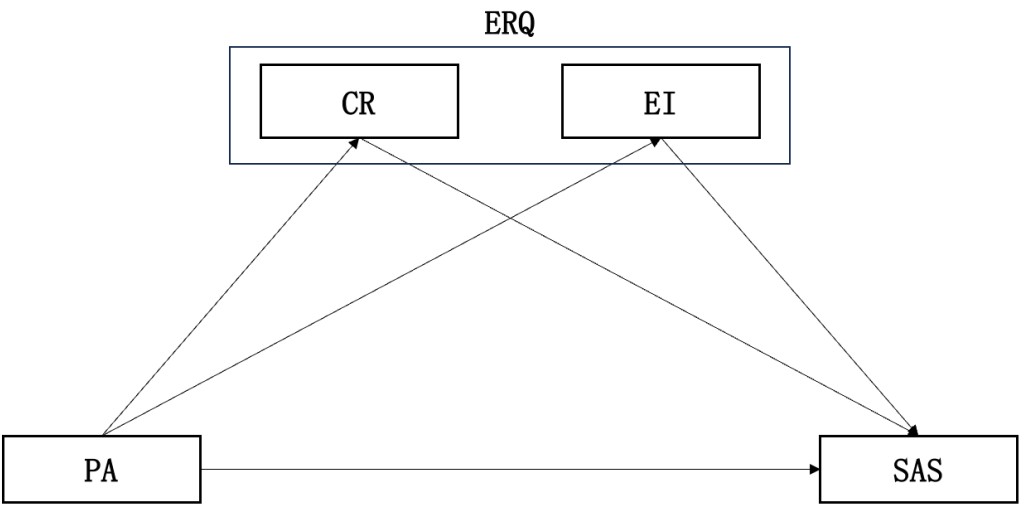

**Figure 1 Mediating regression model of physical activity, anxiety level and emotion regulation among college students.** Abbreviation: PA, physical activity; SAS, anxiety; ERQ, emotion regulation; CR, cognition reappraisal; EI, expression inhibition.

neither have been studies that have explained the roles between the three factors mentioned and the overall effects. Physical activity improves emotion regulation and anxiety from a physiological mechanism perspective by regulating the hypothalamic-pituitary-adrenal axis. Psychological mechanisms suggest that physical activity can enhance cognitive function, and self-efficacy, and reduce anxiety sensitivity (*Szuhany, Bugatti & Otto, 2015*), which is consistent with the self-efficacy hypothesis (*Artino Jr, 2012*). Additionally, based on Bandura's self-regulation theory (*Bandura, 1991*), emotion self-regulation is reflected in the use of automatic or intentional strategies to generate, maintain, adjust, or express one's emotions. Based on these hypotheses and theories, an integrated model can be proposed to explain the relationships between physical activity, anxiety, and emotion regulation. By investigating the anxiety levels of college students and exploring the effects of physical activity and emotion regulation on the anxiety levels of college students as well as the mediating role mechanism, this study can provide a basis for alleviating the anxiety of the college students from a multidimensional perspective. On this basis, this study takes college students as the research object and adopts a process-oriented approach to test the direct relationship of physical activity and anxiety and how emotion regulation affects the relationship indirectly. Two main research questions will be clarified in this study: (1) Is there a positive relationship between physical activity and anxiety among college students in China? (2) Does emotion regulation play a mediating role in the relationship between physical activity and anxiety? The following hypotheses were proposed to conduct the study by using the method outlined by *Hayes & Matthes (2009)*. In order to clearly identify the effect differences of emotion regulation strategies playing a role in the physical activity-anxiety relationship, model hypotheses were constructed as shown in Fig. 1.

Hypothesis 1 (H1): Physical activity is a negative predictor of anxiety.

Hypothesis 2 (H2): Physical activity is a positive predictor of emotion regulation.

Hypothesis 3 (H3): Emotion regulation is a negative predictor of anxiety.

Hypothesis 4 (H4): Emotion regulation plays a mediating role in the indirect relational effects of physical activity on anxiety.

Hypothesis 5 (H5): Both cognitive reappraisal and expressive inhibition play a mediating role in the indirect relational effects of physical activity on anxiety.

## METHOD

### Participants and survey procedures

Based on the sample size calculation for complex mediation models proposed by *Schoemann, Boulton & Short (2017)*, and referring to the correlation coefficients and mean standard deviations of variables from previous studies, the path coefficients $\alpha$, $\beta$, and c' were set at 0.125, −0.118, and −0.087, respectively. The results indicated that to achieve a statistical test capacity of 0.8, the minimum sample size required is 874, which this study meets.

Convenience sampling is a commonly used non-probability sampling method in educational and social science research, with advantages such as low workload, low cost, minimal time investment, and simple operation. However, this method has certain limitations. Distributing questionnaires at different locations to achieve diversity can help mitigate these drawbacks, thereby capturing the typical characteristics of the target population (*Golzar, Tajik & Noor, 2022*). Convenience sampling method was used, and East China, South China, and Central China were selected as the survey areas. The questionnaire survey was conducted in October 2021 and undergraduates of six universities in Shanghai, Jiangsu, Shandong, Guangxi, and Hunan were selected as respondents. The inclusion criteria of the subjects were no mental illness, no use of psychotropic drugs, no contraindications to exercise, no history of head trauma, and no consumption of caffeinated and alcoholic beverages within 24 h of filling in the questionnaire. A total of 1,721 questionnaires were distributed in this study, 84 invalid questionnaires were excluded after verification, 1,637 valid questionnaires were taken back, and the validity rate of the questionnaires was 95.12%.

A questionnaire was used in this study. The questionnaires were distributed by trained principals and collected online and with the assistance of course lecturers. Guided by uniform prompts, students are supervised in the classroom to use cell phones or computers to complete the questionnaires. The recovery rate of the questionnaires was 100%. After the questionnaires were collected, further preliminary screening of the quality of the questionnaires was conducted and the following exclusion criteria were set to obtain a valid questionnaire: (1) Patterned responses: questionnaires where responses exhibit artificial patterns, such as repetitive or consistent answers like "wave-like" or "parallel"; (2) missing data: questionnaires with more than one-third of the responses missing; (3) identical answers: questionnaires where all responses are completely identical. These criteria are set to ensure the validity of the collected questionnaires.

This study was approved by the Ethics Committee of Shanghai University of Sports (102772022RT114). All college students participated voluntarily, and all researchers

involved in data collection, data analysis, and manuscript writing were familiarized with the survey procedures and received thorough ethics training prior to data collection.

## Measuring tools
### Physical activity rating scale-3
The physical activity rating scale-3 (PARS-3) was revised by *Liang (1994)* from Wuhan Sports University and has good reliability and validity with a retest reliability of 0.82 (*Ren & Li, 2020*). The PARS-3 is applicable to the survey of physical activity among youth groups in China (*Liang, 1994*). The scale mainly investigates the participants' physical activity in the last month. It includes the intensity, time, frequency, and program of the physical activity, in which the activity is graded by intensity, time, and frequency, and the activity =intensity × time × frequency. Scoring criteria are as follows: the intensity and frequency of exercise are divided into five grades, with 1–5 points respectively; the time of exercise is also divided into five grades, with 0–4 points; 19 points or less (including 19 points) are for low-intensity physical activity, 20–42 points are for medium-intensity physical activity, and 43 points or more (including 43 points) are for high-intensity physical activity; the maximum total score is not more than 100 points, and the minimum total score is not less than 0 points.

### Self-rating anxiety scale
The self-rating anxiety scale (SAS) was developed by *Zung (1971)*, a Chinese American professor, and is a common scale for diagnosing anxiety (*Zung, 1971*), which can tap into the specific nature of anxiety symptoms (*Dunstan & Scott, 2020*). The Chinese version of the scale has shown good validity and reliability among Chinese, with a Cronbach $\alpha$ coefficient of 0.98 (*Huang et al., 2022*). It has been widely used in the self-assessment of college students' anxiety levels. The scale consists of 20 items, of which 15 items were designed to assess negative emotions (*e.g.*, "I feel scared for no reason" and "I feel like I might be going crazy") and physical conditions (*e.g.*, "I feel that my arms and legs are shaking and trembling" and "I feel my heart beating fast"). Five items (the 5th, 9th, 13th, 17th, and 19th items) were self-assessments of positive emotions (*e.g.*, "I fall asleep easily and sleep well through the night."). Our scoring criteria are shown in the *Wen et al. (2023)*, with higher scores indicating more obvious anxiety symptoms.

### Emotion regulation questionnaire
The emotion regulation questionnaire (ERQ) was developed by *Gross & John (2003)*. The ERQ has two dimensions: cognitive reappraisal and expressive inhibition, and consists of 10 items, with four items measuring the expressive inhibition dimension and six items measuring the cognitive reappraisal dimension. There were six items of cognitive reappraisal dimensions (*e.g.*, "when I am faced with a stressful situation, I make myself think in a way that helps me stay calm"; and "when I want to feel more positive emotions, I change the way I think about the situation"). The Chinese version has been previously validated with good validity and reliability, with Cronbach's $\alpha$ coefficients of 0.85 and 0.77 for the cognitive reappraisal and expressive inhibition dimensions, respectively (*Gross & John, 2003*; *Li et al., 2007*). The ERQ uses a 7-level scoring method, with "1" indicating strongly

disapprove, "2" indicating rather disapprove, "3" indicating somewhat disapprove, "4" indicating uncertain, "5" indicating somewhat agree, "6" indicating rather agree, and "7" indicating strongly agree. Scoring principle are as follows: the scores of the two dimensions were calculated separately, with higher scores indicating a higher percentage of use. A higher or lower score indicates the differences in the individual's use of the two emotion regulation strategies, cognitive reappraisal and expressive inhibition.

## Data analysis

Data analysis were conducted by adopting SPSS 24.0. K-S non parametric test was used to test the normality of the data, one-way ANOVA test was used to verify the homoscedasticity of the data, and Pearson Correlation Analysis was performed. The data were tested to be in conformity with normal distribution and satisfied the requirement of homoscedasticity and linearity. Furthermore, as this study adopted a questionnaire self-report format, this may lead to common method biases. Harman's single-factor test was used to verify the possible common method bias, and the results showed that the first factor explained 19.99% of the total variance, which is less than the critical value of 40%. Therefore, there was no significant common method bias in this study.

After the questionnaire, SPSS 24.0 was used for statistical analysis. Firstly, descriptive statistics and independent samples $t$-tests were performed on the scores of the two dimensions of anxiety level and emotion regulation of college students respectively. Secondly, one-way multiple analysis of covariance was used to control for BMI, place of student source and other relevant demographic variables, to discuss the relationship between grade, physical activity and the two dimensions of anxiety level and emotion regulation of college students respectively. Lastly, mediating effect regression model Process Model 4 was used (M is said to be a mediating variable between X and Y or M plays a mediating role between X and Y if the effect of the independent variable X on the dependent variable Y is studied through a variable M) to control demographic variables (gender and grade) and to analyze the relationship between physical activity, anxiety level and emotion regulation (*Demming, Jahn & Boztug, 2017*). The statistical significance criterion was set based on whether the Bootstrap confidence interval (CI) of 5,000 repeat samples included zero. If the CI did not include zero, the effect was considered significant.

## RESULTS

### Demographic characteristics

The survey included a total of 1,637 university students, comprising 749 males (45.8%) and 888 females (54.2%), with an age range of 18 to 22 years old. Among them, 1,562 students were from the Han nationality (93.2%), and 111 students were from ethnic minorities (6.8%). A total of 822 students (50.2%) were from urban areas, while 815 students (49.8%) were from rural areas.

### Basic characteristics of anxiety levels and emotion regulation in college students

The results in Table 1 show that the overall anxiety level of college students is close to the dividing value for anxiety symptoms, with a total score of (44.72 ± 10.37); and emotion

**Table 1 Comparison of anxiety levels and emotion regulation scores among college students ($x \pm s$).**

| Classification indicators | | Statistics | Anxiety level | Cognition reappraisal | Expression inhibition |
|---|---|---|---|---|---|
| Total | | | $44.72 \pm 10.37$ | $30.16 \pm 6.51$ | $16.96 \pm 4.99$ |
| Gender | Male | | $44.99 \pm 11.21$ | $30.24 \pm 7.00$ | $17.86 \pm 5.09$ |
| | Female | | $44.48 \pm 9.60$ | $30.09 \pm 6.08$ | $16.20 \pm 4.77$ |
| | | $t$ value | 0.985 | 0.468 | 6.788 |
| | | $p$ value | 0.325 | 0.640 | <0.0001 |
| Grade | Freshman | | $41.54 \pm 10.74$ | $31.93 \pm 7.17$ | $16.93 \pm 5.82$ |
| | Sophomore | | $44.78 \pm 10.14$ | $30.01 \pm 6.34$ | $16.88 \pm 4.91$ |
| | Junior | | $46.61 \pm 10.94$ | $29.47 \pm 6.68$ | $17.24 \pm 4.96$ |
| | Senior | | $45.33 \pm 10.21$ | $30.50 \pm 6.68$ | $17.32 \pm 4.53$ |
| | | $F$ value | 7.649 | 5.092 | 0.488 |
| | | $p$ value | 0.0001 | 0.002 | 0.690 |
| Place of student source | Urban | | $44.45 \pm 10.62$ | $30.41 \pm 6.66$ | $16.94 \pm 5.13$ |
| | Rural | | $44.99 \pm 10.10$ | $29.91 \pm 6.36$ | $16.98 \pm 4.84$ |
| | | $t$ value | $-1.059$ | 1.544 | $-0.162$ |
| | | $p$ value | 0.290 | 0.123 | 0.871 |

regulation is at a moderately low level, with a cognitive reappraisal score of ($30.16 \pm 6.51$) and an expression inhibition score of ($16.96 \pm 4.99$).

The results of one-way ANOVA showed that from the perspective of gender, the score of male students' anxiety level were slightly higher than that of female students' and male students' cognitive reappraisal level was slightly lower than female students', neither of which was significantly different ($P > 0.05$). Male students' expression inhibition level was significantly higher than female students', and there was a significant difference ($P < 0.05$). From the perspective of grades, the anxiety level was the highest in junior grade and the lowest in the freshman grade, and there was a significant difference between grades ($F = 7.649$, $P < 0.01$). The cognitive reappraisal level was the highest in the freshman grade and the lowest in the junior grade, and there was a significant difference between grades ($F = 5.092$, $P < 0.01$). However, there was no significant differences in the level of expression inhibition ($P > 0.05$). Regarding the place of student source, there were no significant differences in anxiety levels, cognitive reappraisal and expression inhibition between urban and rural college students ($P > 0.05$), indicating that family conditions are not the main factor influencing college students' anxiety levels and emotion regulation.

## Correlation analysis of physical activity, anxiety level and emotion regulation of college students

To test the direct effects of physical activity and emotion regulation on college students' anxiety levels, bivariate Pearson correlation analysis were conducted for each of the three variables. The correlation coefficients are shown in Table 2.

As can be seen from Table 2, there is a significant correlation between the level of physical activity and anxiety level and the two dimensions of emotion regulation ($r = 0.128$

**Table 2  Correlation between physical activity, anxiety level and emotion regulation among college students.**

| Variables | 1 | 2 | 3 | 4 | 5 |
|---|---|---|---|---|---|
| 1. Physical activity level | 1 | | | | |
| 2. Anxiety level | −0.087[**] | 1 | | | |
| 3. Emotion regulation | 0.125[**] | −0.118[**] | 1 | | |
| 4. Cognitive reappraisal | 0.128[**] | −0.263[**] | 0.886[**] | 1 | |
| 5. Expression inhibition | 0.078[**] | 0.112[**] | 0.796[**] | 0.426[**] | 1 |

Notes.

[**]Represents $p < 0.01$.

$\sim 0.078$, $P < 0.01$). There is also a significant correlation between the two dimensions of emotion regulation and anxiety level ($r = 0.112 \sim 0.263$, $P < 0.01$). There was a high degree of correlation between physical activity, anxiety level, and emotion regulation among college students, and they interacted with each other. It indicates that the data are suitable for mediation model test. In addition, the correlation between the three variables were consistent with theoretical expectations, which provides preliminary evidence for subsequent test of the hypotheses.

## The effect of physical activity on anxiety level and emotion regulation of college students

As can be seen in Table 3, the differences between anxiety level and emotion regulation and multiple back testing were statistically significant in terms of physical activity. Significant differences were observed in anxiety levels and physical activity levels ($F = 6.102$, $P < 0.01$), with the highest level of anxiety in light exercise and the lowest in hard exercise. The effect of light exercise on anxiety levels is greater than that of hard exercise When it comes to the two dimensions of emotion regulation, significant differences were observed in the level of cognitive reappraisal ($F = 9.376$, $P < 0.001$), with the lowest level of cognitive reappraisal in light exercise and the highest in hard exercise. Significant differences were also observed in the level of expression inhibition ($F = 4.089$, $P < 0.05$), with the lowest expression inhibition levels for medium exercise and the highest for hard exercise. Based on back testing of significant effects, the impact of light exercise on the cognitive reappraisal dimension is smaller than that of medium exercise. Medium exercise has a smaller effect on the cognitive reappraisal dimension compared to hard exercise. Moreover, medium exercise has a greater effect on the expressive suppression dimension compared to hard exercise. Hard exercise consistently shows greater effects on the overall dimension of emotion regulation compared to both light and medium exercise. The results suggest that hard exercise is more favorable for college students to ease anxiety and improve the level of emotion regulation.

## Mediation model test of physical activity affecting college students' anxiety level through emotion regulation

In this study, the mediating effect regression model was constructed with the level of physical activity as the independent variable, the level of anxiety as the dependent variable, and emotion regulation as the mediating variable. In the mediation model of physical

**Table 3 Differences in anxiety and emotion regulation levels among college students at different levels of exercise (x ±s).**

| Variables | Light exercise | Moderate exercise | Hard exercise | F value | back testing |
|---|---|---|---|---|---|
| Anxiety level | 45.36 ± 10.14 | 44.54 ± 10.61 | 43.09 ± 10.60 | 6.102** | 1>3 |
| Emotion regulation | 46.49 ± 9.78 | 47.14 ± 8.99 | 48.90 ± 10.19 | 7.69*** | 1<3, 2<3 |
| Cognitive reappraisal | 29.61 ± 6.51 | 30.58 ± 6.26 | 31.29 ± 6.61 | 9.376*** | 1<2, 1<3 |
| Expression inhibition | 16.88 ± 4.90 | 16.56 ± 5.10 | 17.6 ± 5.06 | 4.089* | 2<3 |

**Notes.**
*represents $P < 0.05$.
**represents $P < 0.01$.
***represents $P < 0.001$.
1= light exercise, 2= medium exercise, 3= hard exercise.

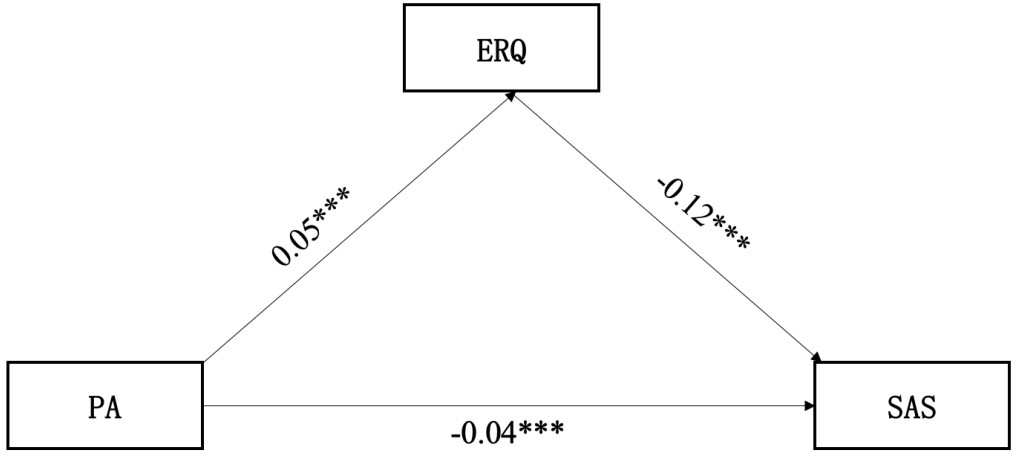

Note: *** represents $p < 0.001$

**Figure 2 Mediating regression model of physical activity, anxiety level and emotion regulation among college students.** Abbreviation: PA, physical activity; SAS, anxiety; ERQ, emotion regulation. Note: *** represents $p < 0.001$.

activity, anxiety level, and emotion regulation, the test results from the model constructed in Fig. 2 and Table 4 showed that physical activity had a direct negative effect on anxiety level, which means the greater the amount of physical activity, the lower the anxiety level ($\beta = -0.042$ $P < 0.001$). Hypothesis 1 thus held. Physical activity had a significant positive effect on the overall level of emotion regulation ($\beta = 0.05$, $P < 0.001$), and hypothesis 2 thus held. Emotion regulation has a negative predictive effect on anxiety level, *i.e.,* the higher the level of emotion regulation, the lower the anxiety level ($\beta = -0.12$, $P < 0.001$). Hypothesis 3 thus held. Based on the combination of hypotheses 2 and 3, the above results indicate that emotion regulation has a mediating role between physical activity and the anxiety level of college students, so hypothesis 4 held. Emotion regulation plays a mediating role in the relationship between physical activity and anxiety level.

**Table 4** Mediation model test for physical activity, anxiety level, and emotion regulation among college students.

| Predictors | ERQ | | | SAS | | | SAS | | |
|---|---|---|---|---|---|---|---|---|---|
| | $\beta$ | SE | $t$ | $\beta$ | SE | $t$ | $\beta$ | SE | $t$ |
| Gender | −1.150 | 0.51 | −2.27 | −1.31 | 0.54 | −2.44 | −1.18 | 0.54 | −2.19 |
| Grade | −2.83 | 0.35 | −0.81 | 1.30 | 0.37 | 3.51 | 1.33 | 0.37 | 3.58 |
| PA | 0.05 | 0.01 | 4.11 | −0.04 | 0.01 | −3.59 | −0.05 | 0.01 | −4.04 |
| ERQ | | | | −0.12 | 0.03 | −4.51 | | | |
| $R^2$ | | 0.14 | | | 0.18 | | | 0.14 | |
| $\triangle R^2$ | | 0.01 | | | 0.03 | | | 0.02 | |
| $F$ | | 10.65*** | | | 12.80*** | | | 10.16*** | |

Notes.
***Represents $P < 0.001$.

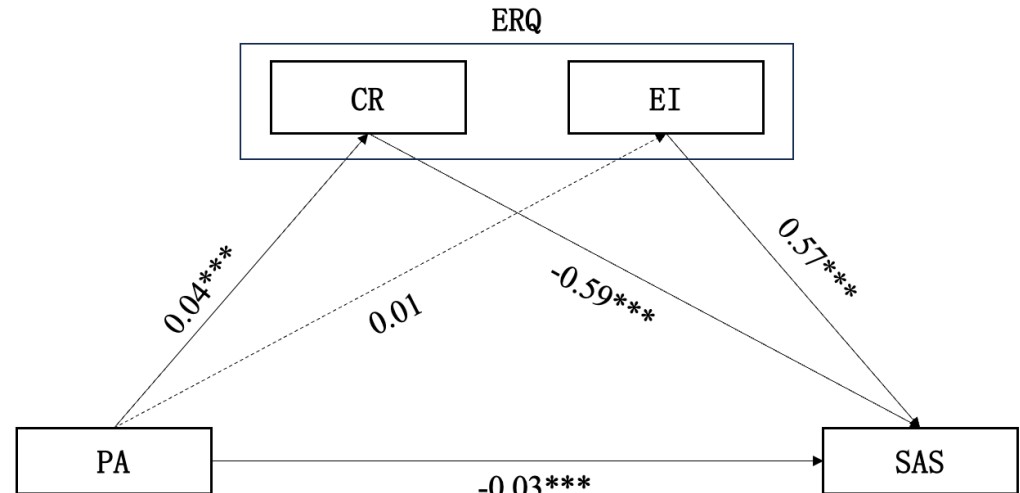

Note: *** represents $p < 0.001$

**Figure 3** **Mediating regression model of physical activity, anxiety level, cognitive reappraisal and expression inhibition among college students.** Abbreviation: PA, physical activity; SAS, anxiety; ERQ, emotion regulation; CR, cognition reappraisal; EI, expression inhibition. Note: *** represents $p < 0.001$.

In the mediation model analysis of physical activity, anxiety level, cognitive reappraisal and expression inhibition (as shown in Fig. 3 and Table 5), the effect of physical activity on emotion regulation can be concluded as follows: physical activity had a significant positive effect on cognitive reappraisal ($R^2 = 0.14$, $\beta = 0.04$, $P < 0.001$), while physical activity did not have a significant effect on expression inhibition ($R^2 = 0.17$, $\beta = 0.01$, $P = 0.27$). The effects of physical activity and emotion regulation on anxiety are as follows: physical activity ($\beta = −0.0.03$, $P = 0.012$), cognitive reappraisal ($\beta = −0.59$, $P < 0.001$), and expressive inhibition ($\beta = 0.57$, $P < 0.001$) had a significant effect ($R^2 = 0.37$) on anxiety levels. The model effect relationship is that the direct and indirect effects of physical activity on anxiety level were −0.745 and −0.354, respectively.

**Table 5   Mediation model test for physical activity, anxiety level, cognitive reappraisal, and expressive inhibition among college students.**

| Predictors | CR | | | EI | | | SAS | | |
|---|---|---|---|---|---|---|---|---|---|
| | $\beta$ | SE | t | $\beta$ | SE | t | $\beta$ | SE | t |
| Gender | 0.41 | 0.34 | 1.22 | −1.57 | 0.26 | −6.11 | | | |
| Grade | −4.45 | 0.24 | −1.91 | 0.16 | 0.18 | 0.91 | | | |
| PA | 0.04 | 0.01 | 5.32 | 0.01 | 0.01 | 1.11 | −0.03 | 0.01 | −2.51 |
| CR | | | | | | | −0.59 | 0.04 | −14.29 |
| EI | | | | | | | 0.57 | 0.05 | 10.51 |
| $R^2$ | | 0.14 | | | 0.17 | | | 0.37 | |
| $\triangle R^2$ | | 0.02 | | | 0.03 | | | 0.14 | |
| F | | 10.83*** | | | 16.03*** | | | 52.45*** | |

Notes.
***Represents $P < 0.001$.

**Table 6   Mediating effect test of physical activity, anxiety levels and emotion regulation among college students.**

| | Variables | $\beta$ | SE | 95% confidence interval | |
|---|---|---|---|---|---|
| | | | | LLCI | ULCI |
| Direct effect | Physical activity → anxiety level | −0.028 | 0.011 | −0.049 | −0.006 |
| Indirect effect | Cognitive reappraisal, expression inhibition →anxiety level | −0.019 | 0.114 | −0.028 | −0.012 |
| | Cognitive reappraisal → anxiety Levels | −0.023 | 0.005 | −0.032 | −0.014 |
| | Expression inhibition → anxiety level | 0.003 | 0.003 | −0.003 | 0.010 |

As can be seen in Table 6, physical activity has an indirect negative effect on anxiety levels by affecting cognitive reappraisal, *i.e.,* the greater the amount of physical activity, the higher the level of cognitive reappraisal ($\beta = 0.4$, $P < 0.001$), and the lower the level of anxiety ($\beta = -0.003$, $P < 0.001$). However, physical activity does not have an effect on expression inhibition ($\beta = 0.01$, $P > 0.05$); expression inhibition had a direct effect on anxiety levels, and the lower the level of expression inhibition, the lower the anxiety level ($\beta = 0.57$, $P < 0.001$). therefore, hypothesis 5 could not hold.

# DISCUSSION

To construct a model of the mediating effect of physical activity on college students' anxiety and emotion regulation. Physical activity was used as the predictor variable, emotion regulation as the mediating variable, and anxiety as the outcome variable to test the relationship between these variables. The results showed that physical activity had a significant positive predictive effect on anxiety. Emotion regulation had a mediating effect, of which cognitive reappraisal played a mediating role.

## Anxiety levels and emotion regulation of college students

The results of the study show that the anxiety level among college students differs in terms of grade and gender, which is similar to what previous studies have indicated (*Naceanceno et al., 2022*). In terms of gender, previous studies have shown that female students have higher anxiety levels than male students do (*Gao, Ping & Liu, 2020*), while the results of

this study shows that male students score slightly higher than female students do in anxiety levels. The potential reason for this may be due to the fact that male students face more pressure from finances, family burdens, employment, *etc.* when they are about to graduate. In terms of grade level, the anxiety level is the lowest in freshman year, the highest in junior year, and the second highest in senior year. The reason for this phenomenon may lie in the following: freshmen feel freshness and joy about the new environment, and they are in a state of relaxation, so it is normal that the anxiety level is the lowest in this period. The highest anxiety level appears in the third year of college, indicating that the anxiety state of college students has already been ahead of schedule. With the rapid development of the society, the expansion of colleges and universities, and the increase in the pressure of talent competition, the prospect of employment is not optimistic, and have to prepare for the graduation of college, making the anxiety level increase with grades (*Fu & Zhai, 2023*). The result suggests that schools should pay more attention to the mental health of juniors in advance. In terms of place of student source, the difference in anxiety levels between urban and rural college students is not significant, which means that place of student source is not the main factor affecting anxiety levels.

The two dimensions of emotion regulation show that there is no significant difference in the level of cognitive reappraisal between male and female students, which is consistent with the findings of foreign scholars (*Caprara et al., 2008*) However, there is a significant difference between male and female students in expression inhibition. Male students' expression inhibition level is significantly higher than female students', which is due to the fact that male students tend not to express, while female students express more emotions than male students do (*Wang et al., 2013*; *Gross & John, 1995*; *Gross & John, 1997*; *Wang et al., 2019*; *Zhang & Bian, 2020*). This may be related to the fact that family and social environments have different role expectations for male and female students (*Jiang, Zhang & Mao, 2018*). There is a significant difference in cognitive reappraisal between grades, which may be related to the level of anxiety faced by themselves. The anxiety level is the lowest in freshman year. Freshmen are enthusiastic, resulting in high level of cognition reappraisal. The anxiety level is the highest in junior year. Juniors are depressed, resulting in the lowest level of cognitive reappraisal. There is no significant difference in the expression inhibition between grades, which suggests that college students do not express their emotions easily with the increase in grades. easily express their emotions. The place of student source is not a major factor affecting emotion regulation, indicating that there is no difference in emotion regulation among college students regardless of whether they come from rural or urban areas.

## Relationship between physical activity and anxiety

Studies have found that physical activity have a direct negative effect on anxiety levels. The effect was even more significant as the amount of exercise increased and anxiety levels among college students decreased dramatically, which is in line with the results of the study by *Ji et al. (2022)* on the effect of physical activity on the anxiety of college students. The relief of anxiety in the high-intensity exercise group was significantly higher than that in the low-intensity exercise group (*Ji et al., 2022*). The potential reasons why physical

activity is effective in relieving anxiety levels are as follows. Firstly, exercise promotes an increase in brain-derived neurotrophic factor (BDNF), which promotes neuroplasticity, neuronal growth and differentiation (*Ji et al., 2022*). Secondly, exercise stimulates the secretion of endorphins, a chemical in the brain. It is a naturally occurring substance that affects mood and makes people feel exciting, which makes people feel relaxing and joyful, and effectively reduces negative emotions such as anxiety (*Duan et al., 2014*). The larger amount of exercise can facilitate the brain and the whole body's blood circulate, make people feel exciting, which is beneficial to college students to relieve psychological pressure, distract their attention, and thus effectively reduce the anxiety level of college students.

The results of this study are slightly different from other studies. Medium amount of exercise improves and reduces anxiety levels of college students (*Duan et al., 2014*; *Ma, 2017*; *Yu, 2008*). However, medium and small amount of exercise may be more helpful to reduce the anxiety level (*Li, 2005*). This may be related to the way, program, and time of participating in physical activity. It is also related to having exercise habits or not, because different forms of physical activity, such as aerobic exercises, strength exercises, *etc.*; different kinds of exercise programs, such as running, playing ball games, Tai Chi, yoga, *etc.*; and different exercise times and exercise intensities may contribute to different degrees of psychological benefits. There is also an interaction between these influencing factors (*Li, 2005*; *Yu, 2008*). It may also be related to the physical activity rating scale used. The scale used in this study (PARS-3) was revised by Liang Deqing et al. from Wuhan Sports University, and the amount of exercise is divided into grades by intensity, time, and frequency, and the amount of exercise $=$intensity $\times$ time $\times$ frequency. The amount of exercise in the existing studies is divided into light, medium, and high intensity based on the Rating of Perceived Exertion (RPE) scores(*Li, 2005*), or in the form of a self-report scale (*Xie, Zhao & Wang, 2002*). As a result, the degree of quantification of the amount of physical activity may vary. In addition, the physical activity scenario is also an important factor influencing the amount of exercise. A period of physical activity at a medium amount of exercise is effective in relieving anxiety in both the autonomous and cooperative scenarios, but physical activity performed in the cooperative scenario relieves anxiety better than physical activity in the autonomous scenario (*Li, 2013*; *Johnston et al., 2021*).

## Mediating role of emotion regulation

In the relationship between physical activity and the effect of emotion regulation among college students, the amount of the exercise has a significant effect on emotion regulation. With the increase of the amount of exercise, the level of cognitive reappraisal of college students is enhanced. The lowest level of expression inhibition is found in the medium amount of exercise, and the highest level of expression inhibition is found in the large amount of exercise. This suggests that increasing the amount of physical activity moderately can effectively enhance the effect of emotion regulation among college students, and that the medium amount of physical activity is a key factor influencing the effect of emotion regulation among college students, which is consistent with the results of *Xue & Zhao (2018)*. Research in cognitive neuroscience has found that the brain regions activated by aerobic exercise and those of emotion regulation overlap in the ventral and dorsal

prefrontal and anterior cingulate cortex regions (*Etkin, Buechel & Gross, 2015*), revealing the brain-neural mechanisms by which aerobic exercise affects emotion regulation. Thus, physical activity enhances emotion regulation, such as the generation of emotion regulation strategies (*Brand et al., 2018*). In addition, we found that college students with high levels of physical activity frequently use cognitive reappraisal of emotion regulation strategies, which enable them to increase the effectiveness of cognitive reappraisal, and blunt emotional responses to negative emotional cues. It is typically associated with a thicker anterior cingulate cortex in the right hemisphere of the brain (*Rothermund & Koole, 2011*; *Wu et al., 2022*). However, our findings showed that physical activity did not have a significant effect on expression inhibition and previous findings did not show that the thickness of thicker anterior cingulate cortex in the right brain was associated with expression inhibition in college students, which may be due to the fact that the participants were in early adulthood and there was less direct correspondence between their anterior cingulate cortex of right brain and expression inhibition. Based on this, physical activity mainly improves college students' emotion regulation by influencing cognitive reappraisal of emotion regulation strategies and thus improving their emotion regulation.

The present study found that emotion regulation is a negative predictor of anxiety level. In other words, anxious patients engaged in less reappraisal, which is similar to what previous studies have shown. The close connection between the prefrontal cortex, striatum, and amygdala gets mature when emotion regulation is enhanced, which enhances prefrontal control over emotionally reactive subcortical areas, and in turn improves the ability of negative emotion regulation (*Ahmed, Bittencourt-Hewitt & Sebastian, 2015*; *Ernst, 2014*).

The mediating regression model constructed in this study confirms the mediating effect mechanism of emotion regulation between physical activity and college students' anxiety level, *i.e.*, physical activity can not only directly lower the anxiety level of college students, but also indirectly lower it by improving their cognitive reappraisal level, which is similar to the results of the existing research conducted by *Jiang, Zhang & Mao (2018)*. A number of studies have shown that physical activity can play a role through emotion regulation, which can ease anxiety and release repressed emotions, and can effectively reduce or eliminate college students' anxiety (*Hu et al., 2004*; *Lin & Xu, 2020*; *Xu & Xu, 2010*). The results of this study further indicate that physical activity has a significant positive effect on the cognitive reappraisal of college students. Besides, physical activity and cognitive reappraisal have a significant negative effect on the anxiety level of college students, which suggests that the effect of physical activity on the anxiety level of college students is mediated by the cognitive reappraisal of emotion regulation. Therefore, enhancing the emotion regulation ability of college students is an important way to leverage the strength of physical activity on college students' anxiety level, which can not only directly lower the anxiety level by promoting the secretion of endorphins and other hormones through physical activity, but also lower the anxiety level of college students by enhancing their sense of satisfaction and joyful emotional experience in physical activity (*Zhu, Lu & Dong, 2016*).

### Limitations and influences

There are some limitations to this study. Firstly, the study utilized convenience sampling, which may introduce sample biases and systematic errors. Future research should expand to larger-scale surveys to validate the findings of this study. Secondly, subjective self-perception assessments have inherent limitations. Future research suggests that measurements should be made with the help of scientific instruments, such as accelerometers. Lastly, the cross-sectional study design inherently limits establishing causal relationships. Future research should include longitudinal intervention experiments to further investigate causal relationships between variables.

Overall, this study theoretically supports the existing research on the relationship between physical activity and college students' anxiety and enriches the relevant theoretical support on anxiety research. In addition, the constructed mediating regression model of physical activity, anxiety, and emotion regulation reveals the interaction between the three and provides an empirical framework. These findings offer theoretical support for understanding the potential mechanisms underlying the relationship between physical activity and anxiety among college students.

## CONCLUSION

This study explored the impact of physical activity on anxiety levels among college students and its underlying mechanisms. It found that physical activity not only directly influences anxiety levels but also exerts an indirect effect through emotion regulation. The study confirmed that higher levels of physical activity and greater intensity are associated with lower anxiety levels. Emotion regulation partially mediates the relationship between physical activity and anxiety levels, deepening our understanding of how physical activity affects anxiety among college students. This suggests a need to prioritize medium to hard physical activities among college students. Additionally, the dynamic nature of emotion regulation underscores its adaptability to different environments and sources of stress. In university settings, students can benefit from using cognitive reappraisal strategies more frequently to manage and alleviate internal anxiety.

### Funding
The authors received no funding for this work.

### Competing Interests
The authors declare there are no competing interests.

### Author Contributions
- Xinxin Sheng conceived and designed the experiments, performed the experiments, analyzed the data, prepared figures and/or tables, authored or reviewed drafts of the article, and approved the final draft.

- Xili Wen performed the experiments, analyzed the data, prepared figures and/or tables, authored or reviewed drafts of the article, and approved the final draft.
- Jiangshan Liu performed the experiments, authored or reviewed drafts of the article, and approved the final draft.
- Xiuxiu Zhou analyzed the data, authored or reviewed drafts of the article, and approved the final draft.
- Kai Li conceived and designed the experiments, performed the experiments, analyzed the data, prepared figures and/or tables, authored or reviewed drafts of the article, and approved the final draft.

### Human Ethics

The following information was supplied relating to ethical approvals (*i.e.*, approving body and any reference numbers):

Shanghai University of Sports granted ethical approval to carry out the study within its facilities (Ethical Application Ref: 102772022RT114).

### Field Study Permissions

The following information was supplied relating to field study approvals (*i.e.*, approving body and any reference numbers):

This study was approved by the Ethics Committee of Shanghai University of Sports (102772022RT114).

### Data Availability

The raw measurements are available in the Supplementary File.

### Supplemental Information

Supplemental information for this article can be found online at http://dx.doi.org/10.7717/peerj.17961#supplemental-information.

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
