# Peer review of "Effects of physical activity on anxiety levels in college students: mediating role of emotion regulation"

_PeerJ, doi:10.7717/peerj.17961_

## Round 0.1 · original submission · Major Revisions

The study entitled Effects of Physical Activity on Anxiety Levels in College Students: Mediating Role of Emotion Regulation” demonstrated excellent findings using an appropriate methodological approach. However, some important points must be clarified in the manuscript. Your article has great potential for publication on PeerJ, but the reviewers have requested substantial changes to be made in text.

·

Basic reporting

1. Overall, this is an interesting study, but the authors still need to further clarify the practical significance of this study.
2.Emotion regulation is a multidimensional structure that includes different strategies of emotion regulation(in line 92-93), What are their roles? Obviously, hypothesis 5 of the study is not well supported.

Experimental design

1. The establishment of the mediation model may require sample size calculation, please add. 2. The method of subject selection still needs to be further supplemented. Are there any differences among subjects sampled in multiple regions? 3. The proportion of sophomores is significantly higher. Does this affect the extrapolability of the conclusion? At the very least, comparisons between grades may be inappropriate, please revise. 4. The study examined too few demographic characteristics to adequately describe the characteristics of the selected subjects for comparison in subsequent studies. 5. The reliability and validity test of the scale should not refer to previous literature, but need to be calculated by the author. 6. Is the scale collected online or offline? Or a combination of the two? Their quality control seems to be difficult to uniform, please explain this. In addition, the author needs to supplement how to eliminate outliers and invalid values from the scale data.

Validity of the findings

1. Multiple group comparisons should be supplemented by multiple post-hoc comparisons. 2. Table 4 and Table 5 seem to show the results of regression analysis rather than the results of the intermediary model. Please check whether they are wrong.

Reviewer 2 ·

Basic reporting

The study "Effects of Physical Activity on Anxiety Levels in College Students: Mediating Role of Emotion Regulation" addresses an important topic. The manuscript's structure and language are adequate, but there are instances where clarity is compromised. Streamline the content to remove redundant information and use more precise language to enhance clarity. There are significant concerns regarding its methodology, analysis, and interpretation of results that necessitate a major revision.
Introduction
The introduction, while sufficiently establishing the study's background, does not adequately tie the current literature to the study's specific objectives. There is a lot of the latest literature available with scientific findings. The authors should rewrite the introduction section with the newest literature.

I’m unable to find the operationalization of the study's proposed hypotheses. Simply writing hypotheses is not enough.

Experimental design

The most significant concern lies in the study's sampling method. The use of convenience sampling from specific regions raises questions about the generalizability of the findings to the broader population of college students. Additionally, the design is cross-sectional, which limits the ability to infer causality between physical activity, anxiety levels, and emotion regulation. Provide a more detailed description of the methodology, including the rationale for choosing specific measurement tools and their validation status. Include a thorough discussion of the ethical considerations and steps taken to ensure data privacy and informed consent.

Validity of the findings

The analysis lacks depth in exploring potential confounding factors that could influence the relationship between physical activity, anxiety, and emotion regulation. The study’s current findings, while interesting, do not provide a compelling advancement in understanding the complex interplay between physical activity and mental health in college students. The study presents a positive bias, focusing primarily on results that support the hypothesis while under-discussing contradictory or non-significant findings. The statistical analysis, while comprehensive, lacks a discussion on the potential impact of confounding variables. Enhance the statistical analysis section by including considerations of possible confounders and their effect on the study's results.
The discussion around the implications of findings is cursory and does not sufficiently consider the limitations of the data or alternative explanations. It raises concerns about the strength and validity of the conclusions drawn. The manuscript would greatly benefit from a more critical analysis and discussion of the results, including a thorough consideration of the study's limitations and potential biases. Expand the discussion to place the study's findings within a broader research context, drawing comparisons with other studies and theoretical frameworks.
The conclusion section in its current form is totally irrelevant and doesn’t reflect the main crux of the findings. Restrict conclusions to what the data directly supports, particularly avoiding causal interpretations from correlational data. The study's limitations are not adequately explored, which could give an overestimated impression of its applicability. There is a lack of forward-looking statements, such as suggestions for future research or practical applications of the findings.

Reviewer 3 ·

Basic reporting

The purpose of this article was to explore the effects of physical activity on college students' anxiety levels and to test the mediating role of emotion regulation in this context. The authors used 1,721 college students from five regions, namely, Shanghai, Jiangsu, Shandong, Guangxi, and Hunan, as study subjects, who were assessed and investigated by the Physical Activity Rating Scale (PARS-3), the Anxiety Self-Rating Scale (ERQ), and the Emotion Regulation Scale (ERQ).

During my review, I noted that the authors provided a detailed description of the study population, instruments and methods, and that the article was clearly structured in line with the requirements of academic writing. However, I believe that there are still some deficiencies in some areas, as described below:
(1)In the introductory part of the study, the focus should be on analyzing why the variable of emotion regulation(erq) was chosen. This is because current research has been clearer about the dangers of anxiety and the relationship between PA and anxiety.
(2)There is no elaboration based on the appropriate theory in the preamble section, i.e., under what theoretical framework the authors have proposed this model and hypotheses. This was not clearly stated, leaving the reader unclear as to how the model was proposed and what issues or theories the hypotheses were based on. If this issue is not well articulated, then the rest of the study does not make very much sense.
(3)The study design, despite the description of convenience sampling in the limitations section, is not clear about how the convenience sampling was carried out and the quality control of the included subjects.
(4)In the conclusions section of the study, it is recommended to add a description of what the results of this study mean for practice.

Experimental design

It is recommended that the authors add the following: Why is convenience sampling used? If random sampling is not possible, how exactly is convenience sampling conducted and how is quality controlled?

Validity of the findings

Suggested author: Since it is convenient to sample, the expression of significance in the results should be cautious

---

## Round 0.2 · Minor Revisions

The study entitled “Effects of Physical Activity on Anxiety Levels in College Students: Mediating Role of Emotion Regulation” demonstrated interesting findings using an appropriate methodological approach. However, very minor revisions must be clarified in the manuscript.

Your article has great potential for publication on PeerJ.

·

Basic reporting

The revised manuscript has basically met the requirements.

Experimental design

Authors should also supplement the inclusion and exclusion process for participants.

Validity of the findings

No comment

Additional comments

References need to be formatted. In addition, some recent references should be added.

Reviewer 2 ·

Basic reporting

The authors have addressed my all comments. The overall quality of the manuscript has been improved and accept for publication in the current form.

Experimental design

No comment

Validity of the findings

No comment

---

## Round 0.3 · accepted · Accept

Dear Author,

Congratulations, after the good work of revisions in response to the reviewers' comments, I would like to inform you that your manuscript has been accepted for publication in PeerJ.